# Prediction of the Spontaneous Estrus Expression Period Based on Large (≥10 mm) Follicle Numbers in Lactating Holstein Dairy Cows

**DOI:** 10.3390/vetsci10030231

**Published:** 2023-03-17

**Authors:** Ryotaro Miura, Takuma Inoue, Yuka Kunugi, Miya Yasukawa, Saku Koyama, Rena Sato, Tomochika Matsumura, Tsuyoshi Tajima, Itaru Yoshimura, Tadaharu Ajito

**Affiliations:** 1Faculty of Veterinary Science, School of Veterinary, Nippon Veterinary and Life Science University, Kyonan, Musashino, Tokyo 180-8602, Japan; 2Fuji Animal Farm, Nippon Veterinary and Life Science University, Fujikawaguchiko, Yamanashi 401-0338, Japan

**Keywords:** dairy cow, follicle dynamics, large follicle, estrus prediction, ultrasonography

## Abstract

**Simple Summary:**

We evaluated the estrus prediction method focusing on ≥10 mm follicle (large follicle) numbers with functional corpus luteum in lactating Holstein dairy cows. We divided cows into two groups with one and two or more large follicles at the ovarian examination using transrectal ultrasonography. Estrus detection was performed after an ovarian examination. More than 75% of estrus occurred 10 days after the ovarian examination in cows with one large follicle at the ovarian examination. Whereas more than 75% of estrus occurred within 9 days of the ovarian examination in cows with two or more large follicles at the ovarian examination. Hence, we propose that the evaluation of ≥10 mm follicle numbers could be useful to predict the estrus expression period in lactating Holstein dairy cows.

**Abstract:**

The objectives of this study were (1) to investigate the distribution of large (≥10 mm) follicle numbers during the estrous cycle and (2) to compare the timing of the estrus expression period after the ovarian examination between cows with one large follicle (1F) and two or more large follicles (2F) with functional corpus luteum (CL) at the ovarian examination in lactating Holstein dairy cows. In experiment 1, we performed 393 ovarian examinations by ultrasonography, addressed the existence of CL (≥20 mm) and large follicle numbers, and classified cows into 1F (*n* = 229) and 2F (*n* = 164) groups. The 1F appearance rates were beyond 75% each day during 3 to 12 d after estrus. However, 2F appearance rates were beyond 75% each day during 15 to 24 d after estrus. In experiment 2, we performed 302 ovarian examinations by ultrasonography and classified cows into the 1F (*n* = 168) and 2F (*n* = 134) groups. Estrus detection was performed for 24 d after the ovarian examination in each cow. In the 2F group, 75% of estrus occurred within 9 d of the ovarian examination. However, 75% of estrus occurred 10 d after the ovarian examination in 1F. Days from the ovarian examination to estrus were significantly shorter in the 2F (6.0 d; median, 7.2 ± 4.0 d; mean ± SD) than in the 1F (13 d, 12.4 ± 4.3 d) group. In conclusion, focusing on ≥10 mm follicle numbers with CL could be useful for predicting the estrus expression period.

## 1. Introduction

Higher estrus detection rates contribute to higher reproductive performance in dairy farms [1]. Although estrus detection is mainly dependent on the visual observation of estrous behavior performed by dairy farmers, the staff cannot spend enough time on estrous detection due to the increasing size of the herd and the recent lack of well-trained workers in the dairy industry [2]. Therefore, it would be useful for dairy workers to indicate estrus timing predictions to improve estrus detection rates and work efficiency in daily tasks.

Lactating dairy cows have two or three follicular waves during the estrous cycle [3,4]; however, about 80% (78.6 to 83.3%) of lactating dairy cows have two follicular waves [5,6,7]. Follicular wave emergence was defined as several small follicles (around 3.0 mm) starting to develop [8]. The follicular wave that developed soon after estrus was called the first follicular wave (first-wave) [9,10]. During the next 2–3 days, one of the follicles will become the dominant follicle (DF), while the others will become subordinate follicles [11]. From the former studies, when the largest follicle in the first wave reaches 8.5 mm in heifers [12] or 9.1 mm in lactating dairy cows [13], it becomes the DF; in addition, all the largest follicles that reach 10 mm are defined as DF in the experimental design [14], therefore, follicles that are larger than 10 mm could be defined as DF. The first-wave DF develops soon after estrus and undergoes atresia during the first 8 to 12 days of the estrous cycle [15]. When the first-wave DF undergoes atresia, the second follicular wave (second-wave) DF starts to develop. Previous studies showed that the size of the atretic first-wave DF does not diminish promptly during the late estrous cycle [16]. Therefore, the size of the atretic first-wave DF maintains more than 10 mm (large follicle) during the second-wave DF development in lactating dairy cows; in other words, we could have two morphologically present follicles, but both are not physiologically active because one is going to atresia (Figure 1). In addition, although the average large follicle numbers are 1.5 per day during 4 to 12 days from ovulation, these are 2.0 per day during 14 to 20 days from ovulation in lactating dairy cows [6]. Based on these results, the number of large follicles would increase during the late estrous cycle. Taken together, we hypothesized that we could predict the estrus expression timing based on the large follicle numbers. However, no studies are available on the evaluation of estrous prediction based on the large follicle numbers with functional CL in lactating Holstein dairy cows. Furthermore, the distribution of large follicle numbers during the estrous cycle has not been investigated in detail using large replicate numbers.

Therefore, this study aimed (1) to investigate the distribution of follicle (≥10 mm: large follicle) numbers during the estrous cycle and (2) to compare the timing of estrous expression after the ovarian examination between one large follicle and two or more large follicles at the ovarian examination in lactating Holstein dairy cows. This study would offer an alternative estrus prediction method, and this method would be beneficial for the reproductive management of lactating Holstein dairy cows.

## 2. Materials and Methods

### 2.1. Animals, Housing, and Feeding

The study was conducted from June 2016 to March 2019 at the Fuji Animal Farm of Nippon Veterinary and Life Science University, located in Fujikawaguchiko-machi, Yamanashi, Japan. We investigated lactating Holstein dairy cows in this experiment. Cows were managed in a tie-stall barn, and they were milked two times a day, and they had free access to water throughout the experimental period. Cows were fed 13 kg of Timothy hay, 3 kg of alfalfa hay as roughage divided into two times a day, and 4 to 13 kg of concentrate (16% dry matter CP) and 2.0 to 2.5 kg of beet pulp divided into five times a day. Concentrate and beet pulp were provided by an automatic concentrate feeder (Max Feeder HID, Orion).

All the experimental procedures followed the Guidelines for the Care and Use of Animals established by Nippon Veterinary and Life Science University, and all animal protocols were approved by the Institutional Animal Care and Use Committee (Nippon Veterinary and Life Science University, Tokyo, Japan (No. 29J-12, 30J-17)). 

### 2.2. Study Design

We visited the farm once a week, every Saturday, to examine the ovaries. We recorded ovarian structures (CL and follicle) at the ovarian examination of each cow on a weekly basis. Ovarian examinations were performed by ultrasonography (ImaGo veterinary ultrasound scanner, IMV imaging) equipped with a 7.5 MHz linear transducer. We defined the functional CL as ≥20 mm, which size was referred to in the previous studies [17,18] and large follicles as ≥10 mm in diameter, and recorded CL and large follicle numbers. Three well-trained farm staff members observed cows for the detection of spontaneous estrus by visual observations, including swelling or hyperemia of the vagina, mucous from the vagina, flehmen, and bellowing twice a day. When farm staffs detected the estrus, an artificial inseminator confirmed the preovulatory follicle and regressed CL by rectal palpation to evaluate the estrus. If the estrus was confirmed, we recorded the day of estrus. We performed two evaluations described as follows: experiment 1, investigating the distribution of large follicle numbers during the estrous cycle, and experiment 2, comparing the timing of estrous expression after ovarian examination between the one large follicle (1F) and two or more large follicles (2F) groups.

In experiment 1, we investigated 419 ovarian examinations (days in milk: 170 ± 103, parity: 2.8 ± 1.4, milk yield: 34.7 ± 7.7 kg/day; mean ± SD) from 78 cows at the weekly farm visit. We calculated the days from the latest estrus to the ovarian examination in each cow that had functional CL and at least one large follicle. We examined cows with the latest estrus detected before 0 to 24 days from the ovarian examination. Numbers and appearance rates of 1F and 2F were calculated each day from the estrus. For example, see the Experiment 1 part of Figure 2. We did ovarian examinations of cows 1, 2, and 3 at 0 d, and we recorded the size and number of CL and large follicles in each cow. If the cows had CL≥20 mm and at least one follicle with 10 mm, we checked the day of the latest previous estrus in each cow. When estruses were confirmed 12, 6, and 3 days before ovarian examination in cows 1, 2, and 3, respectively, we determined the days of the estrous cycle as 12, 6, and 3 days in cows 1, 2, and 3, respectively. Then, we evaluate the relationship between days of the estrous cycle and large follicle numbers (1F vs. 2F). 

In experiment 2, we investigated 331 ovarian examinations (days in milk: 164 ± 111, parity: 2.9 ± 1.4, milk yield: 35.4 ± 7.6 kg/day) from 68 cows at the weekly visit to the farm. We evaluated the estrus for 24 days after the ovarian examination in each cow that had a functional CL and at least one large follicle. Moreover, we calculated the days from the ovarian examination to the estrus in each cow. Estrus numbers and appearance rates after the ovarian examinations were calculated for each day of 1F and 2F, respectively. For example, see the Experiment 2 part of Figure 2. We did ovarian examinations of cows 4, 5, and 6 at 0 d, and we recorded the size and number of CL and large follicles in each cow. If the cows had CL ≥20 mm and at least one follicle with 10 mm, we would wait for the day of next estrus in each cow after the ovarian examination day. When estruses were detected 14, 9, and 4 days after ovarian examination in cows 4, 5, and 6, respectively, we evaluated the days between ovarian examination and the next estrus expression day in each cow. Then, we evaluated the relationship between large follicle numbers (1F vs. 2F) at the ovarian examination and the number of days from the ovarian examination to estrous expression. 

Figure 2 shows a schematic diagram of the experimental model.

### 2.3. Statistical Analysis

All the data of this experiment were analyzed using EZR (version 1.54, Saitama Medical Center, Jichi Medical University), the graphical user interface for R (the R Foundation for Statistical Computing). More precisely, it is a modified version of R commander designed to add statistical functions frequently used in biostatistics [19].

Days from the latest estrus to the ovarian examination and from the ovarian examination to the estrus between 1F and 2F were analyzed using mixed-effect multiple linear regression. Days from the latest estrus to the ovarian examination and from the ovarian examination to the estrus were analyzed as dependent variables. The independent variables were large follicle number (1F, 2F), parity (primiparous, multiparous), and milk production (<35 kg, ≥35 kg), and the interactions between large follicle number and each independent variable were incorporated into the linear regression model as fixed effects. Milk production was measured every month on this farm. We used the values obtained in the month when the cows underwent the ovarian examination. Milk production showed a normal distribution in this study; thus, we classified each independent variable into two groups by the mean (35.0 kg). In all analyses, cows were included as a random effect. The *p* values <0.05 were considered statistically significant.

## 3. Results

### 3.1. Animals

In experiment 1, we excluded five ovarian examinations without large follicles and functional CL, two ovarian examinations without large follicles, and nineteen ovarian examinations without functional CL from the analysis. As a result, we classified the ovarian examinations into the 1F (*n* = 229) and 2F (*n* = 164) groups and proceeded with them for the final analysis. The ovarian examinations of primiparous and multiparous cows were 78 and 315, respectively. The ovarian examinations of cows with <35.0 kg and ≥35.0 kg milk per day yielded 211 and 182, respectively.

In experiment 2, we excluded two ovarian examinations without large follicles and functional CL, two ovarian examinations without large follicles, and twenty-five ovarian examinations without functional CL from the analysis. As a result, we classified the ovarian examinations into 1F (*n* = 168) and 2F (*n* = 134) groups and proceeded with them for the final analysis. The ovarian examinations in primiparous and multiparous cows were 55 and 247, respectively. The ovarian examinations of cows with <35.0 kg and ≥35.0 kg milk per day were 152 and 150, respectively.

### 3.2. Experiment 1: The Effect of Large Follicle Numbers on the Days from the Latest Estrus to the Ovarian Examination, and Distribution of Numbers and Appearance Rates of the Day from the Latest Estrus in the 1F and 2F Groups

The linear regression revealed that a large follicle number (*p* < 0.001) was associated with the days from the latest estrus to the ovarian examination. Parity (*p* = 0.094), milk production (*p* = 0.240), the large follicle number and parity interaction (*p* = 0.105), and the large follicle and milk production interaction (*p* = 0.285) were not associated with the days from the latest estrus to the ovarian examination. 

The days from the latest estrus to the ovarian examination in 2F were longer than those in 1F. Table 1 shows the mean and median days from the latest estrus to the ovarian examination in the 1F and 2F groups.

Figure 3 shows the frequency distribution of numbers (Figure 3a) and appearance rates (Figure 3b) of the day from the latest estrus to the ovarian examination of the 1F and 2F groups. We observed 75.1% of 1F (172 of 229) during the 3 to 12 days after the estrus, and 89.6% of 2F (147 of 164) during the 13 to 24 days after the estrus. The 1F appearance rates were beyond 75% each day during 3 to 12 days after the estrus, while the 2F appearance rates were beyond 75% each day during 15 to 24 days after the estrus, except 20 days after the estrus (2F appearance rate of 60%). In addition, the 1F and 2F appearance rates were approximately 50% on 13 and 14 days after estrus.

### 3.3. Experiment 2: Frequency Distribution of Numbers and Appearance Rates of the Estrus Expression Day from the Ovarian Examination of the 1F and 2F Groups, and the Effect of Large Follicle Numbers on Estrus Expression Timing after the Ovarian Examination

The linear regression revealed that a large follicle number (*p* < 0.001) associated with the days from the ovarian examination to the estrus. The days from the ovarian examination to the onset of estrus in 2F were shorter than those in 1F. Parity (*p* = 0.060), milk production (*p* = 0.404), the large follicle number and parity interaction (*p* = 0.279), and the large follicle and milk production interaction (*p* = 0.479) were not associated with the days from the latest estrus to the ovarian examination.

The days from the ovarian examination to the onset of estrus in 2F were shorter than those in 1F. Table 2 shows the mean and median days from the ovarian examination to the onset of estrus in the 1F and 2F groups.

Figure 4 shows the frequency distribution of numbers (Figure 4a) and appearance rates (Figure 4b) of the estrus expression day from the ovarian examination of the 1F and 2F groups. We observed that 76.1% (102 of 134) estruses occurred within 9 days of the ovarian examination in the 2F group, while 78.0% (131 of 168) estruses occurred after 10 days of the ovarian examination in the 1F group.

## 4. Discussion

In this study, we evaluated the distribution of large follicle numbers during the estrous cycle and estrus timing after ovarian examination between one large follicle and two or more large follicles with functional CL at the ovarian examination in lactating Holstein dairy cows. The present study showed that (1) the majority of the cows during the early estrous cycle had only one large follicle, and their majority during the late estrous cycle had two or more large follicles, and (2) the majority of the cows with one large follicle expressed estrus after 10 days from the ovarian examination. In contrast, the majority of the cows with two or more large follicles expressed estrus within 9 days of the ovarian examination.

A previous study showed that although the average large follicle number is 1.5 per day during 4 to 12 days from ovulation, these numbers are 2.0 per day during 14 to 20 days from ovulation in lactating dairy cows [6]. This result means that the 1 to 2 large follicles were observed during the early- to mid-estrous cycle and two during the mid- to late-estrous cycle. Because the first-wave DF size does not diminish promptly after atresia [16], large follicles that exist during the mid- to late-estrous cycle consists of the atretic first-wave and second-wave DFs. As expected, the majority of 1F appeared within 12 days of the estrus, and the majority of 2F appeared after 13 days from the estrus in this study.

Follicle development dynamics were affected by plasma progesterone during the estrous cycle [20,21,22]. Moreover, plasma progesterone is affected by daily milk production [23,24]. In addition, energy status and milk production are different between primiparous and multiparous cows [25]. We speculated that the appearance of 1F and 2F might be affected by milk production level and parity. However, based on the multiple linear regression analysis, no significant effect of the large follicle number and parity interaction or the large follicle number and milk production interaction could be observed on the days from the latest estrus to the ovarian examination. Therefore, we could evaluate the state of the estrous cycle using large follicle numbers by ultrasonography without considering milk production level or parity.

In previous studies, estrous cycle stages were estimated by the visual characteristics of the CL (size, color, and vascularity) and the existence of >10 mm follicles [26] or pixel values of ultrasound CL images [27]. However, in the former study, we had to evaluate the ovaries of a live cow by examination under direct visual inspection. Therefore, this method could not be applied for practical use. In the latter study, the CL pixel values of the early and mid-estrous cycles were not different. Therefore, evaluating the state of the estrous cycle during the luteal phase might be difficult using this method. In the present study, we proposed an alternative method for estrous cycle stage (3 to 12 days vs. >13 days) estimation using large follicle numbers with functional CL.

Based on the above-described results, we evaluated the timing of estrus expression after the ovarian examination. More than 75% of estrus was observed within 9 days from the ovarian examination in 1F, and more than 75% of estrus was observed after 10 days from the ovarian examination (Figure 4). In addition, we did not observe significant effects of the interaction of large follicle number and parity or large follicles and milk production on the days from the ovarian examination to the estrus. Therefore, we could apply this evaluation regardless of milk production and parity in lactating Holstein dairy cows. Figure 5 presents a schematic diagram of the simplified model of the estrus expression timing mechanism. From the practical aspect, we could offer useful information to the veterinarian for predicting the estrus expression period for dairy farmers. In addition, timed AI programs require approximately 10 days until AI from the initial treatment [28,29,30]. Therefore, if we predicted the estrus expression within 9 or 10 days, we could judge whether to use a timed AI program for earlier AI or not. In addition, in the case of applying the Ovsynch protocol, it is recommended that Ovsynch treatment be initiated during the early luteal phase (between 5 and 12 days of the estrous cycle) [31,32,33]. Therefore, if we focused on the functional CL and large follicle numbers, we could select the applicable cases of the Ovsynch protocol for lactating dairy cows. To target initiation of the timed AI protocol, this estrus prediction method based on large follicle numbers could be useful from a practical standpoint.

However, we should acknowledge the limitations of this study. We could apply this method only to cows with functional CL. Certain cows have unusual ovarian dynamics, and we could not apply this method to them. For example, certain cows showed later atretic timing of the first-wave DF after 14 days of the estrous cycle, or because the first-wave DF diminished less than 10 mm, we observed only one large follicle. Further research would be required for estrus prediction methods of higher accuracy, including the information of DF and CL size and uterus status. 

## 5. Conclusions

We showed that one large follicle was observed more often during the early estrous cycle and two or more large follicles were observed more frequently during the late estrous cycle. More than 75% of estruses occurred within 9 days of the ovarian examination with two or more large follicles, while more than 75% occurred 10 days after the ovarian examination with only one large follicle. Our results suggest that focusing on ≥10 mm follicle numbers with functional CL could be a useful method for predicting the estrus expression period. This information would be beneficial for the reproductive management of lactating Holstein dairy cows.

## Figures and Tables

**Figure 1 vetsci-10-00231-f001:**
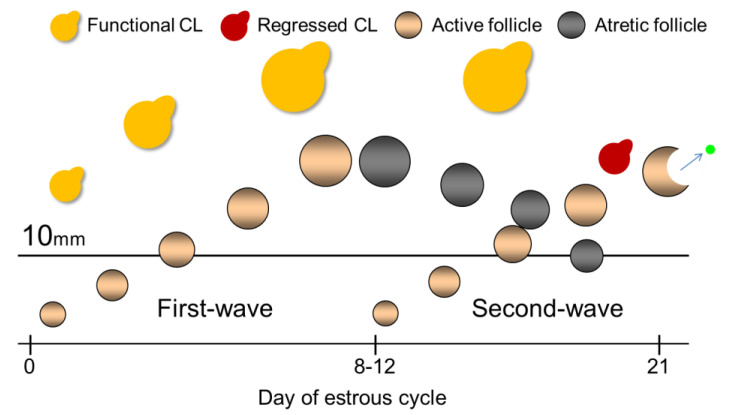
Schematic diagram of the dynamics of ovarian structures (corpus luteum (CL), first-wave dominant follicle (DF), second-wave DF) through the estrous cycle in two-wave pattern cows. The day of estrus is defined as 0 d. It is important to focus on the size of the atretic first-wave DF, which maintains more than 10 mm (a large follicle) during the second-wave DF development during the late estrous cycle.

**Figure 2 vetsci-10-00231-f002:**
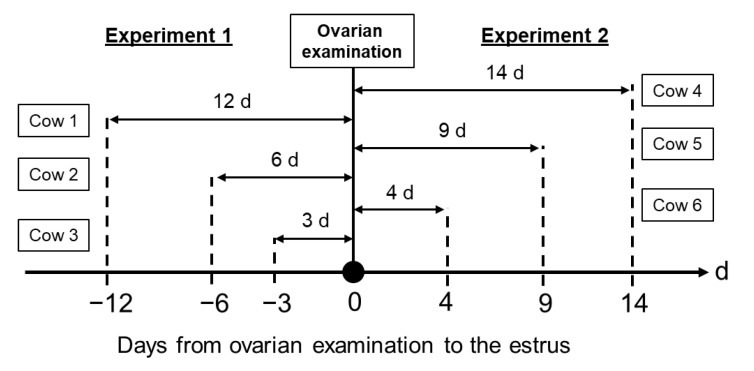
Schematic diagram of the experimental model. We visited the farm once a week, every Saturday, to examine the ovaries in each cow. In experiment 1, the day of the latest estrus was recorded in each cow, and the days from the latest estrus to ovarian examination day (0 d) were calculated. For example, see the Experiment 1 part of Figure 2. We did ovarian examinations on cows 1, 2, and 3 at 0 d, and we recorded the size and number of CL and large follicles in each cow. If the cows had CL ≥20 mm and at least one follicle with 10 mm, we checked the day of the latest previous estrus in each cow. When estruses were confirmed 12, 6, and 3 days before ovarian examination in cows 1, 2, and 3, respectively, we determined the days of the estrous cycle as 12, 6, and 3 days in cows 1, 2, and 3, respectively. In experiment 2, the day of the estrus during the 24 days after the ovarian examination was recorded in each cow, and the days from the ovarian examination day to the estrus were calculated. For example, see the Experiment 2 part of Figure 2. We did ovarian examinations of cows 4, 5, and 6 at 0 d, and we recorded the size and number of CL and large follicles in each cow. If the cows had CL ≥20 mm and at least one follicle with 10 mm, we would wait for the day of next estrus in each cow after the ovarian examination day. When estruses were detected 14, 9, and 4 days after ovarian examination in cows 4, 5, and 6, respectively, we evaluated the days between ovarian examination and the next estrus expression day in each cow. Ovarian examinations were performed by ultrasonography equipped with a 7.5 MHz linear transducer. We defined the functional corpus luteum (CL) as ≥ 20 mm and the large follicle as ≥ 10 mm, and we recorded the functional CL and large follicle numbers on the ovarian examination day.

**Figure 3 vetsci-10-00231-f003:**
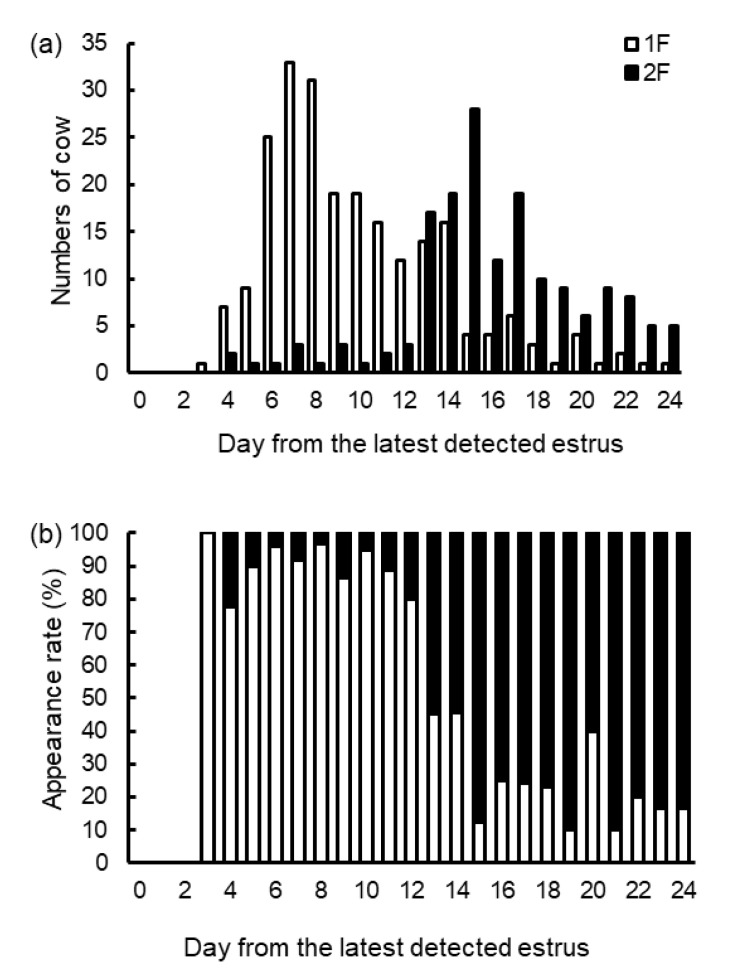
Distributions of numbers and appearance rates of 1F and 2F after the latest detected estrus. The distribution day of the number of cows for each of 1F and 2F is shown in (**a**). The distribution day of the respective appearance rates of 1F and 2F within each day is shown in (**b**). Here, 1F = cows with one large follicle at the ovarian examination, *n* = 229; 2F = cows with two or more large follicles at the ovarian examination, *n* = 164. The day of the latest detected estrus is defined as 0 d. Ovarian examination was conducted once a week, every Saturday, in each cow.

**Figure 4 vetsci-10-00231-f004:**
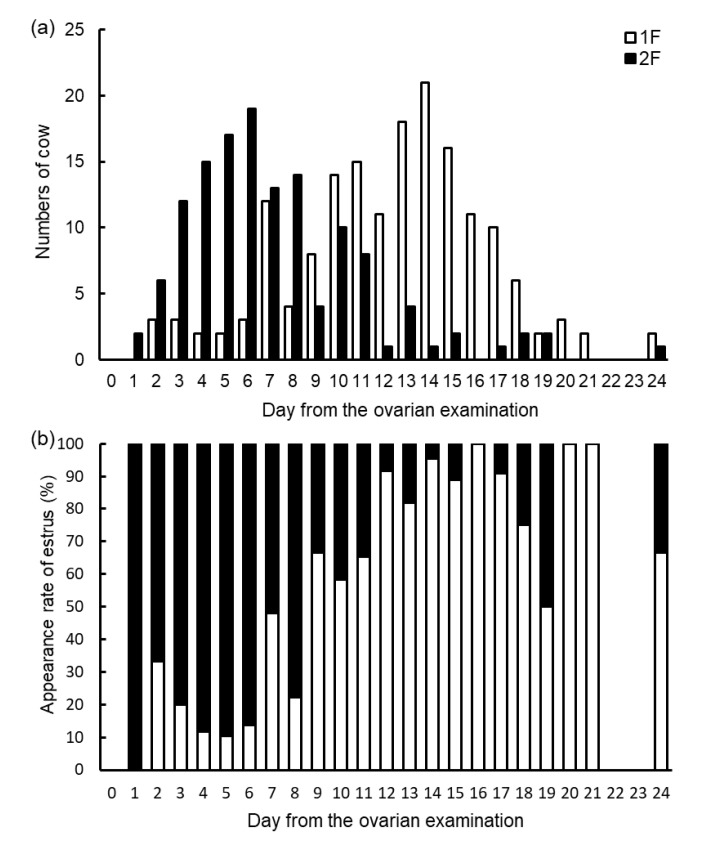
Distribution of numbers and appearance rates of detected estrus in 1F and 2F after the ovarian examination, respectively. The distribution day of the number of detected estruses in 1F and 2F after the ovarian examination is shown in (**a**). The distribution day of the respective appearance rates of detected estrus in 1F and 2F after the ovarian examination within each day is shown in (**b**). 1F = cows with one large follicle at the ovarian examination, *n* = 168; 2F = cows with two or more large follicles at the ovarian examination, *n* = 134. The day of the ovarian examination is defined as 0 d. Ovarian examination was conducted once a week, every Saturday, in each cow.

**Figure 5 vetsci-10-00231-f005:**
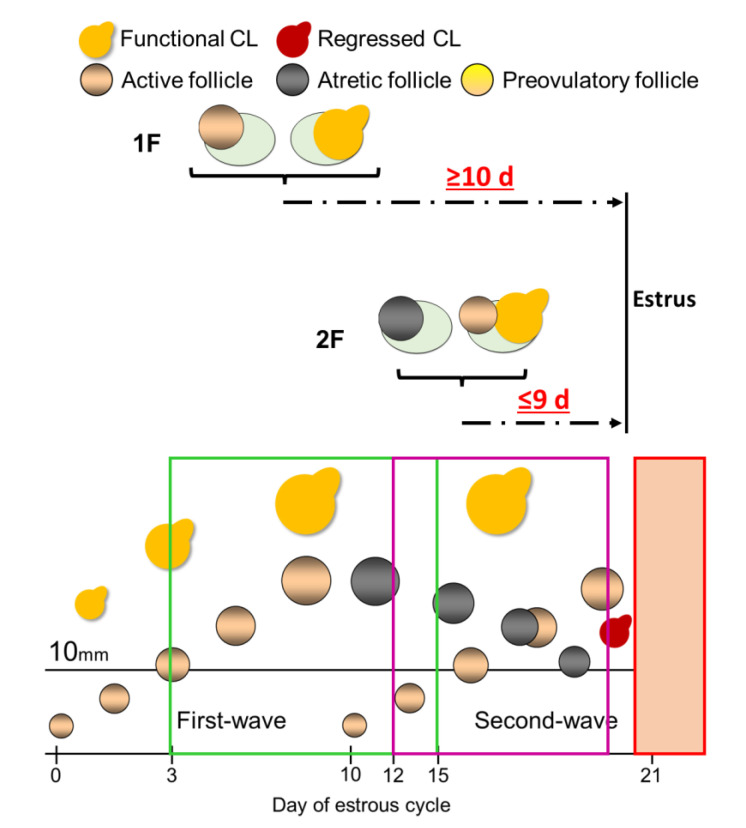
Schematic diagram of a simplified model of the mechanism of estrus expression timing based on the results of the present study. The majority of the cows during the early estrous cycle (from 3 to 12 days after estrus) had only one large follicle, which was composed of the physiologically active first-wave DF, and their majority during the late estrous cycle (15 days after estrus) had two or more large follicles, which were composed of the atretic first-wave DF and the physiologically active second-wave DF. Therefore, the majority of the cows with one large follicle, which were in an early estrous cycle, expressed estrus after 10 days from the ovarian examination. In contrast, the majority of the cows with two or more large follicles, which were in a later estrous cycle, expressed estrus within 9 days of the ovarian examination. Here, 1F = cows with one large follicle at the ovarian examination, *n* = 168; 2F = cows with two or more large follicles at the ovarian examination. A large follicle was defined as having a diameter of ≥10 mm.

**Table 1 vetsci-10-00231-t001:** Differences in mean and median days from the latest estrus to the ovarian examination between 1F ^1^ and 2F ^2^.

End Point	Large Follicle Number (*n*)		*p*-Value
	1F ^1^ (229)	2F ^2^(164)	
Mean	10.0 ± 4.1	16.1 ± 4.0	<0.001
Median	9.0	16.0	
(Lower quartile, Upper quartile)	(7.0, 12.0)	(14.0, 19.0)	

^1^ 1F = cows with one large follicle at ovarian examination. ^2^ 2F = cows with two or more large follicles at ovarian examination. Mean data are represented by mean ± SD.

**Table 2 vetsci-10-00231-t002:** Differences in mean and median days from the ovarian examination to the estrus between 1F ^1^ and 2F ^2^.

End Point	Large Follicle Number (*n*)		*p*-Value
	1F ^1^ (168)	2F ^2^ (134)	
Mean ^3^	12.4 ± 4.3	7.2 ± 4.0	<0.001
Median	13.0	6.0	
(Lower quartile, Upper quartile)	(10.0, 15.0)	(4.0, 9.0)	

^1^ 1F = cows with one large follicle at ovarian examination. ^2^ 2F = cows with two or more large follicles at ovarian examination. ^3^ Mean data are represented by mean ± SD.

## Data Availability

The data presented in this study are available on request from the corresponding author. The data are not publicly available due to restrictions by the research group.

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
