# Peer review of "Prediction of the Spontaneous Estrus Expression Period Based on Large (≥10 mm) Follicle Numbers in Lactating Holstein Dairy Cows"

_vetsci, 2023, doi:10.3390/vetsci10030231_

Round 1
Reviewer 1 Report (Previous Reviewer 1)
The Materials and Methods section has been much improved. However, once this point has been clarified I find some paragraphs of the discussion confusing:
In Lines 244–248 authors stat: “no significant effect of the large follicle number and parity interaction and the large follicle number and milk production interaction could be observed on the days from the latest estrus to the ovarian examination. Therefore, we could evaluate the state of the estrous cycle using large follicle numbers by ultrasonography without considering milk production level and parity” However, in Lines 261–263 claim: “we could observe significant effects of the interaction of large follicle number and parity or large follicle and milk production on the days from the ovarian examination to the estrus” Please clarify
Line 252: “we had to evaluate the ovary by direct visual observation” direct visual observation?
Lines 258–261 do not match with Table 2. Please clarify.
Author Response
Dear Reviewer 1
Thank you for your comments to our manuscript. We read your comments carefully and modified the manuscript. We highlighted the modified part of the manuscript which was suggested by comments of Reviewers and Editor. Please confirm the manuscript. We describe our response to your comments one by one. In addition, we add Editor's comments and our response. Please confirm them.
The Materials and Methods section has been much improved. However, once this point has been clarified I find some paragraphs of the discussion confusing:
AU:
Thank you for your comments. We modified some points of “Materials and Methods”, Please confirm it.
In Lines 244–248 authors stat: “no significant effect of the large follicle number and parity interaction and the large follicle number and milk production interaction could be observed on the days from the latest estrus to the ovarian examination. Therefore, we could evaluate the state of the estrous cycle using large follicle numbers by ultrasonography without considering milk production level and parity” However, in Lines 261–263 claim: “we could observe significant effects of the interaction of large follicle number and parity or large follicle and milk production on the days from the ovarian examination to the estrus” Please clarify
AU:
Thank you for your suggestion. We are sorry for our mistake. “We did not observe” is correct. We changed sentence as below.
L297-299: “In addition, we did not observe significant effects of the interaction of large follicle number and parity or large follicle and milk production on the days from the ovarian examination to the estrus.”
Line 252: “we had to evaluate the ovary by direct visual observation” direct visual observation?
AU:
Thank you for your indication. In this sentence, we want to say that we have to see the ovary directly for estimate the estrous cycle using this reference methods. We changed sentence as below.
L287-288: “However, in the former study, we had to evaluate ovaries of live cow by examination under directly visual inspection.”
Lines 258–261 do not match with Table 2. Please clarify.
AU:
Thank you for your indication. Your indication is correct, Figure 4 is correct, not Table 2. We changed to “(Table 2)” to “(Figure 4)”.
Editor’s comment
- Please revise on the typeset manuscript.
All revisions need to be made on the manuscript that has already been
typeset. Please see attachment for the typeset manuscript.
AU:
Thank you for your comments. We revised manuscript in the typeset. Revised parts were colored as red and yellow. Please confirm it.
- Please increase the word count of the manuscript text during the revision
process. We checked that the text count of your manuscript is 3255. The suggested
minimum word count is 4000 words. Please expand the text count of the article
to more than 4000.
AU:
Thank you for your comments. We expand the text words count more than 4,000.
- Please reduce the similarity rate.
Typically, we require reference single similarity rates to be reduced to less
than 5%. Attached is the similarity report for the manuscript vetsci-2262919.
The report shows that the single citation rate of reference 1 is 12%. Please
Reduce the similarity rate of your manuscript to the standard range.
AU:
Thank you for your recommendation. We modified the manuscript and reduced the similarity rates.
- Please provide Simple Summary.
The journal decided to include "Simple Summary" for the published papers.
"Simple Summary" is supposed to be written for a lay audience, which is very
helpful for paper promotion. Thus, please provide the "Simple Summary" when
you revise. You can refer to the introduction below:
AU’:
Thank you for your indication. We are sorry for attaching simple summary. We add the simple summary in the revised version of manuscript

Reviewer 2 Report (Previous Reviewer 3)
Thnaks to have accept my proposals of changes
Author Response
Dear Reviewer 2
Thank you for your comments to our manuscript. We highlighted the modified part of the manuscript which was suggested by comments of Reviewers and Editor. Please confirm the manuscript. We describe our response to your comments one by one. In addition, we add Editor's comments and our response. Please confirm them.
Thanks to have accept my proposals of changes
AU:
Thank you for your kind comments, suggestion, and indication
Editor’s comment
- Please revise on the typeset manuscript.
All revisions need to be made on the manuscript that has already been
typeset. Please see attachment for the typeset manuscript.
AU:
Thank you for your comments. We revised manuscript in the typeset. Revised parts were colored as red and yellow. Please confirm it.
- Please increase the word count of the manuscript text during the revision
process. We checked that the text count of your manuscript is 3255. The suggested
minimum word count is 4000 words. Please expand the text count of the article
to more than 4000.
AU:
Thank you for your comments. We expand the text words count more than 4,000.
- Please reduce the similarity rate.
Typically, we require reference single similarity rates to be reduced to less
than 5%. Attached is the similarity report for the manuscript vetsci-2262919.
The report shows that the single citation rate of reference 1 is 12%. Please
Reduce the similarity rate of your manuscript to the standard range.
AU:
Thank you for your recommendation. We add the references, and we reduce the similarity rates less than 5.0%.
- Please provide Simple Summary.
The journal decided to include "Simple Summary" for the published papers.
"Simple Summary" is supposed to be written for a lay audience, which is very
helpful for paper promotion. Thus, please provide the "Simple Summary" when
you revise. You can refer to the introduction below:
AU’:
Thank you for your indication. We are sorry for attaching simple summary. We add the simple summary in the revised version of manuscript.

Round 2
Reviewer 1 Report (Previous Reviewer 1)
The manuscript has been revised correctly and improved. I would like to thank the authors for considering my comments and suggestions
This manuscript is a resubmission of an earlier submission. The following is a list of the peer review reports and author responses from that submission.
Round 1
Reviewer 1 Report
This is an interesting study investigating prediction of the spontaneous estrus based on the presence of large follicles (≥ 10 mm) in dairy cows. However, methods are not clear, particularly regarding two key points of the study design:
In both objectives, days post-estrus at the time of ovarian exams should be clearly described. More cows with two large follicles can be observed in the second half of the luteal phase as authors suggest (L55) and show in Figure 3. Therefore, it is obvious that a cows with two follicles on Day 15 of the estrous cycle can show estrus earlier than cows with a single follicle on Day 10. When a corpus luteum was considered to be functional? This point should be clarified throughout the manuscript.
Authors visited the farm weekly (L96). It seems that authors examined ovaries every day (Lines 110–121 and legend of Figure 2, and Figure 3). Please clarify.
Finally, a good prediction of estrus is the previous estrus. This point should be added in the Lines 277–283.
Minor point:
Reference 1- Please add or change to more recent reviews.
Reviewer 2 Report
1.The English Wring is poor and very confusing.
2. In Study Designe part, experiment 1 included 419 oavrian examinations from 78 cows, and experiment 2 included 331 ovarian exminations from 68 cows. That means totally 146 cows were used in this study. But in results part, experiment 1 classified 393 cows in to F1 group(229) and F2 group(164), apart from 26 cows excusion. Experment 2 classified 302 cows into F1 (168) and F2 (134) droups. Why the number is much larger than the total number of cows in Study Design? I suppose the 393 and 302 should be the number of oavrian examinations but not cows.
3. 419 oavrian examinations from 78 cows and 331 ovarian exminations from 68 cows means that approximately 5 ovarian examinations for each cow in average. What are the details for this examinations in the population?
4. No details were described about the time of ovulation and the relationship between ovulation and appearance of large follicles in these cows.
Besides these, there are still many points of problems which I don't list one by one.
Reviewer 3 Report
Congratulations for this very practical paper. The results can offer to the practionner an usefull method to better manage the reproduction. Such approach is also an other application of ultrasound use.
42 : This sentence « therefore, most of lactating dairy cows have two follicular waves » can be suppressed
55 you are right but you can insist to write that you can have two morphological present follicles but both are not physiologically active because one is going to atresia.
57 : you can mention that the possibility to have two dominant follicles during the first or the second wave of the cycle is growing due to the increase of multiple ovulations and so twinning in dairy cattle
61 follicle numbers between 1 and more than 2 : 1 means 1 or two because you analyze more than 2 ?
65 Figure 1 : thanks for this very nice figure
99 During the first 5 days of the œstrus cycle, are you sur that the diameter of the corpus luteum is higher than 20 mm ?
103 bellowing is better than roaring (a cow is not a lion…)
104 farm staff i.e. a vet I suppose
108 in the two experiments, you compare two groups of cows. The first group has 1 or 2 follicles and the group 2 has more than two follicles : is it right ?
122 In the figure 2 coud you explain why you mention cow 1 to cow 6. Moreover you follow the cows during 24 days after an ovarian examination but in the figure you mention only 14 days
123 very nice figure
220 again the difference between 1 large follicle and more than 2 large follicles is not very understandable : what about the situation if only 2 follicles are detected ?
258 we could observe significant effects but in line 168 you wrote « Parity (P = 0.094), milk 166 production (P = 0.240), the large follicle number and parity interaction (P = 0.105), and the 167 large follicle and milk production interaction (P = 0.285) were not associated with the days 168 from the latest estrus to the ovarian examination ». Could you explain ?
263 we could offer to the veterinarian
Reviewer 4 Report
The study by Miura et al., aims, to the understanding of this reader, to improve fertility management of lactating dairy cows by improving the prediction of the time of estrous signs for use in AI, by considering the number of Dominant Follicle (1 vd 2 DF) in the corse of ultrasound monitoring of the ovaries day by day. Although the ingenuity at work by the authors has to be acknowledged, this reviewer definitely fails to understand the significance of this study with regard to improving the fertility of lactating herds. The concept of wave emergence, follicle dominance and so forth have very well been acquired by the scientific community and veterinarians at work in farms. In addition, some data presented by the authors are by far expected (1 DF at the early stages of estrous cycle vs 2 DF at later stages) and have already been reported by a number of previous studies. Finally, from the personal interpretation of this reviewer, it is difficult to imagine how the presented results of this study may improve fertility management of the herd.